# A Novel Hydrothermal CdS with Enhanced Photocatalytic Activity and Photostability for Visible Light Hydrogenation of Azo Bond: Synthesis and Characterization

**DOI:** 10.3390/nano13030413

**Published:** 2023-01-19

**Authors:** Martina Milani, Michele Mazzanti, Giuliana Magnacca, Stefano Caramori, Alessandra Molinari

**Affiliations:** 1Department of Chemical, Pharmaceutical and Agricultural Sciences, University of Ferrara, Via Luigi Borsari 46, 44121 Ferrara, Italy; 2Department of Chemistry, University of Torino, Via P. Giuria 7, 10125 Torino, Italy

**Keywords:** hydrothermal CdS, photocatalysis, photostability, visible light, hydrogenation, azo bond, sulfur vacancies

## Abstract

A good photocatalyst maximizes the absorption of excitation light while reducing the recombination of photogenerated carriers. Among visible light responsive materials, CdS has good carrier transport capacity; however, its photostability is poor and limits its use. Here, the synthesis of a new hydrothermal CdS is reported, and post-synthesis annealing determines crystal properties and spectroscopic characteristics. The introduction of sulfur vacancies as intra band gap states is the key factor for the enhancement of photocatalytic activity. In fact, by spectroscopic and photo-electrochemical experiments, we demonstrate that sulfur vacancies act as an electron sink, favoring the charge transfer process to methyl orange. In addition, the studied hydrothermal CdS is characterized by very high stability, thus enabling a visible-light active photocatalyst that is overall recyclable, stable and more efficient than the commercial benchmark.

## 1. Introduction

Semiconductor photocatalytic technology is undoubtedly of great importance for its potential to solve global energy shortage and environmental pollution. It can use solar energy to produce valuable chemical fuels, such as hydrogen from the photocatalytic splitting of water and hydrocarbons from the reduction of carbon dioxide. Moreover, this technology can be developed to help purify the environment through photocatalytic degradation of various toxic and harmful chemical pollutants [1,2,3,4,5,6,7,8,9].

One of the major difficulties in photocatalysis is how to maximize the absorption of excitation light by photocatalytic materials while reducing the recombination of photogenerated carriers, which depends on the properties of photocatalytic materials, such as the width of the band gap and crystal structure. Titanium dioxide, the traditional semiconductor photocatalyst, has a wide band gap of 3.2 eV, and it can only absorb ultraviolet light (λ < 380 nm), which approximately accounts for only 3–5% of the total solar energy. Therefore, the photocatalytic efficiency of unmodified titanium dioxide under solar light irradiation is severely restricted [5,10,11,12].

Among the various photocatalysts, cadmium sulfide is a visible-light responsive material with a band gap of 2.4 eV. In addition, it has good carrier transport capacity, which can make photogenerated electrons and holes mobile in a timely and efficient manner, making redox processes reliable on its surface, possibly at the expense of fast recombination [13,14]. For these reasons, CdS is considered one of the most prominent semiconductors in photocatalysis [15,16,17]. However, the use of CdS itself presents some fundamental drawbacks, such as poor photostability, accrued from severe photo corrosion and exceedingly fast charge carrier recombination, that limits its efficiency. Practically, chemical stability is a critical matter of concern for comprehending extensive scale application of CdS as a photocatalyst for solar energy conversion [15].

Modification of CdS or its combination with another component is a research hotspot in the field of photocatalytic H_2_ production [15] and Ref. therein [18], and the use of CdS-based photocatalysts for the photocatalytic reduction of CO_2_ gradually has aroused a great interest from researchers in recent years [19,20,21]. They report that there are many factors that affect the efficiency of these reductive processes, including the morphology, crystal structure and composition; in addition, the synthetic method employed plays an important role in establishing and influencing the characteristics of the obtained material [22]. More generally, the location of the conduction band at negative potentials makes CdS a photocatalyst of interest for reductive transformations that take advantage of electrons promoted in the conduction band [23]. For example, we recently demonstrated that FTO/TiO_2_/CdS slides, in which CdS was deposited by chemical bath deposition, were an efficient and stable photocatalytic system for the reductive cleavage of N=N bond in azodyes with visible light [24]. 

Herein, we report about the synthesis of a new CdS-based photocatalyst in which we can enhance the charge transfer process, resulting in a very important photocatalytic activity. The used synthetic method is that of hydrothermal synthesis, which among all the reported preparative procedures, is the most sustainable [15]. We highlight how post-synthesis annealing can determine crystal properties and spectroscopic characteristics of hydrothermal samples and how their synergy affects the photocatalytic performance. As a test reaction we focused on the reductive cleavage of N=N bond in azodyes, a reaction that has been already used for the investigation of the FTO/TiO_2_/CdS junctions [24], considering that this kind of transformation is of interest for depollution, since it transforms a pollutant, widely present in the environment, into useful molecules that can re-enter the production cycle of new dyes [25]. Noteworthy, annealed hydrothermal CdS shows higher photocatalytic activity than commercial CdS powder. In addition, recyclability and high stability have been achieved, leading to a photocatalytic material that overcomes the main shortcomings of CdS.

## 2. Materials and Methods

### 2.1. Chemicals

Cadmium Chloride (CdCl_2_, 99%) was purchased from Alfa Aesar, Thiourea (CH_4_N_2_S, Tu 99%), *L*-Cysteine (C_3_H_7_NO_2_S, 99%) and methyl orange (MO) from Aldrich. The MO structure is reported in Figure 1. All the chemicals were used without further purification. CdS, used as a reference material, was purchased from Aldrich (<5 microns, 99.995% purity).

### 2.2. Hydrothermal Synthesis of Cadmium Sulfide

The photocatalysts were synthesized by hydrothermal reaction [26]. First, 10 mmol CdCl_2_, 16 mmol Tu and 4 mmol *L*-cysteine were dissolved in 80 mL of H_2_O under magnetic stirring. The solution was then transferred into a Teflon-lined stainless autoclave and the hydrothermal reaction was performed at 160 °C for 10 h. At the end, the autoclave was cooled at room temperature. The precipitated powder was filtered and washed several times with H_2_O and ethanol and then dried. Some of the obtained powder was annealed in a muffle at 400 °C for 1 h. Considering the starting reagents (CdCl_2_, thiourea and L-cysteine) the molar ratio Cd:S was 1:2 in the prepared samples. We here referred to the photocatalysts as CdS-HT and CdS-HTa400, where “HT” is for Hydro-Thermal and “a400” is for the Annealed powder at 400 °C.

For photoelectrochemical measurements fluorine doped tin oxide (FTO, Pilkington TEC 7) was used as ohmic support for hydrothermal CdS. Conductive FTO was cleaned via sonication in 2-propanol for 10 min and dried under a warm air stream. The FTO/CdS-HT and FTO/CdS-HTa400 electrodes were prepared by inserting FTO slides, on which was previously spread a ZrO_2_ paste [24], into the autoclave, with the FTO coating facing the bottom of the liner. Deposition on the ohmic support occurred in the same hydrothermal conditions as previously described. Herein ZrO_2_ serves as an electrochemically inert porous support for improving the adhesion of the solvothermal CdS to FTO. For the electrochemical potential involved in this study, ZrO_2_ can indeed be regarded as an insulating substrate. Percolation of charge from the FTO support to the CdS is still possible due to the porous nature of ZrO_2_ film.

The procedure for the preparation of commercial CdS deposited on FTO slides is as follows: 3 g of commercial CdS powder was ground in a porcelain mortar with water (1 mL) and acetylacetone (0.1 mL) to prevent particles reaggregation. After the powder has been well dispersed, it was diluted by adding water (4 mL). Finally, 0.1 mL of Triton X-100 (Aldrich) was added to facilitate the spreading of the colloid on the substrate. The so obtained colloidal suspension was then deposited by blade casting on FTO [27], followed by sintering at 400 °C in air.

### 2.3. Structural, Textural and Morphological Characterization of Photocatalysts

The crystalline phases of the as-synthesized samples were identified by X-ray diffraction using BRUKER D8 Advance X-ray diffractometer equipped with a Sol-X detector, working at 40 kV and 40 mA. The X-ray diffraction patterns were collected in a step-scanning mode with steps of Δ2θ = 0.02° and a counting time of 10 s/step using Cu Kα1 radiation (λ = 1.54056 Å) in the 2θ range of 15–80° using an incident grazing angle set-up. 

The morphologies and the composition of the photocatalysts were examined by a scanning electron microscopy (SEM, Zeiss EVO 40 scanning electron microscope), equipped with an energy-dispersive spectroscopy EDS system (software AZTEC-Oxford). 

Specific surface area and porosity measurements were made using ASAP2020 by Micromeritics as a gas-volumetric instrument for the determination of nitrogen adsorption/desorption isotherms at the temperature of −196 °C. BET model and DFT method (slit pores) were applied to evaluate the exposed surface area and porosity of the materials. Prior to nitrogen adsorption, all the powders were outgassed overnight at the temperature of 80 °C (residual pressure 10^−2^ mbar) in order to remove atmosphere molecules adsorbed onto the surface and into the pores. 

X-ray photoelectron spectroscopy (XPS) was used to examine the chemical composition and chemical states of Cd and S in cadmium sulfide-based materials before and after photocatalytic experiments. Mg Kα excitation was used in combination with a hemispherical analyzer (Phoibos 100, Specs, Berlin, Germany). The survey and high-resolution spectra were acquired with energy resolutions of 1.4 eV and 0.9 eV, respectively, on freshly sputtered silver (Ag 3d). The spectrometer was calibrated to the Au 4f7/2 peak at 84.0 eV. Conductive carbon tape was used to fix and electrically ground the powders. Electrostatic charging effects were observed and corrected by calibrating C 1s to 284.4 eV. The spectra of S 2p and Cd 3d were fitted using the CasaXPS software after Shirley’s background subtraction.

### 2.4. Steady State Optical Absorption and Emission

The UV–visible diffuse reflectance spectra (DRS) of CdS powders were recorded in the range of 400–800 nm, on a Jasco V-570 spectrophotometer equipped with the integrating sphere, using BaSO_4_ as reference. The band gap energy of the catalysts was determined using DRS and applying the Tauc model. Tauc plots were obtained according to (F (R) × hν)^α^ = A × (hν − E_g_), where α = 2 for a direct band gap, (F (R) = (1 − R)^2^/2R) is Kubelka–Munk function, A is a proportionality coefficient and E_g_ is the semiconductor band gap.

Emission spectra (λ_exc_ = 430 nm) of FTO/CdS thin films at room temperature were recorded with an Edinburgh Instruments FLS 920 spectrofluorometer using a dedicated film holder. Slits were set at 5 nm both for emission and excitation during the measurements. Spectra were corrected for the lamp and photomultiplier response and an Oriel 5219 cut off filter was used to remove second order effects. The PL spectra are the average over 10 subsequent scans with a 1 nm step.

### 2.5. Photo Action Spectra

Photoaction spectra were collected under mono-chromatic illumination using a Luxtel 150 W Xe lamp and a photophysics monochromator using a bandwidth of 10 nm. A calibrated silicon photodiode was used to measure incident irradiance before each photocatalytic test. Photoaction spectra were performed between 410 nm and 550 nm. CdS-HTa400 and commercial CdS (3 g/L) were suspended in 3 mL of deaerated aqueous solutions containing MO (10 ppm) and EtOH (10% *v*/*v*) and irradiated for 60 min. After each irradiation the suspension was centrifuged, and the supernatant was collected. The UV–Vis spectrum of the solution was recorded, and the number of MO molecules converted was determined. As a measure of reaction efficiency at each wavelength, formal quantum efficiency (FQE) is used, defined as the number of dye molecules hydrogenated per one incident photon.

### 2.6. Photoelectrochemistry

Photoelectrochemistry characterization of hydrothermal CdS deposited on FTO slides was carried out using Metrohm Autolab PGSTAT 302/N electrochemical workstation under solar simulated illumination with an ABET Sun Simulator (AM 1.5 G filter) whose incident irradiance was set at 0.1 W/cm^−2^ with a Newport model 1918-c Power Meter. The experiments were performed in a three-electrode cell using aqueous 1 M HCOONa as the electrolyte; the reference electrode was SCE; the counter electrode was a Pt wire and FTO/CdS-HT, FTO/CdS-HTa400 or FTO/CdS were used as working electrodes. CV (Cyclic Voltammetry) were performed between 1 and −1 V vs. SCE at a scan rate of 20 mV/s^−1^ both under irradiation and dark conditions. For open circuit chronopotentiometry, initially the prepared thin films were positively polarized in the dark at 0.5 V vs. SCE for 100 s until a stable potential was reached. The substrate was then irradiated with AM1.5 light, generating electrons and holes that may undergo recombination or undergo separation and storage into the semiconductor. The irradiation was maintained until a stable photo-potential was reached. Consequently, restoration of dark conditions causes the decay of photo-potential due to recombination.

### 2.7. Photocatalytic Experiments

Irradiation cycles were carried out with an Oriel Xe/HgXe lamp using a glass cut-off filter (λ > 400 nm). In a typical experiment, the desired amount of photocatalyst powder was dispersed in a quartz cuvette in 3 mL of an aqueous solution containing the dye and EtOH (10% *v*/*v*). The cuvette was closed and degassed by N_2_ bubbling for 30 min. Then, it was placed in front of the lamp and irradiated for the desired period. At the end of the irradiation, the powder was separated by centrifuging the suspension for 10 min at 4000 rpm and the supernatant was recovered.

The optimal amount of photocatalyst was determined by performing several photocatalytic experiments varying the quantity of photocatalyst while keeping the other experimental conditions unchanged. The visible irradiations were performed for 10 min for both CdS-HTa400 and commercial CdS and for 30 min for CdS-HT. The optimal amount so determined was 3 g/L for CdS-HTa400 and the commercial one, while it was 10 g/L for CdS-HT.

The UV–visible spectrum of the irradiated solution was recorded with a Cary 300 UV–Vis double beam spectrophotometer (Agilent Technologies, Santa Clara, CA, USA).

The concentration of the dye was evaluated from the decrease of the absorption maximum at the end of every experiment.

For the recycle experiments after centrifugation, the powder was recovered, washed several times, and employed again in subsequent experiments in the same conditions.

## 3. Results and Discussion

### 3.1. Morphologic, Compositional and Optical Characterization

XRD diffraction patterns of commercial CdS, CdS-HT and of CdS-HTa400 are reported in Figure 1. For CdS-HT (red line) and of CdS-HTa400 (blue line), the patterns can be assigned to the hexagonal wurtzite CdS phase by comparing with the standard data from JCPDS card (File No. 41-1049). The seven main diffraction peaks (located at 2θ angle 24.88°, 26.58°, 28.26°, 36.74°, 43.8°, 47.94° and 51.94°) corresponding to the (100), (002), (101), (102), (110), (103) and (112) planes, respectively, are very similar to the results reported by Srinivasa Rao et al. [28].

It is observed that after annealing at 400 °C, the powder improves its crystallinity. In fact, the intensity of the peaks increases, and widths decrease simultaneously. XRD patterns of hydrothermal samples are different from that of commercial CdS, which shows the presence of both hexagonal and cubic crystalline structures (black line). The average crystallite size (*d*) was calculated using the Debye Scherrer formula:(1)$$d=\frac{0.9\lambda }{\beta cos\theta }$$
where *d* is the particle size, *β* is full width at half maximum (FWHM) in radians of the XRD peaks, θ is the diffraction angle and *λ* (1.54056 Å) is the wavelength of X-ray used. The results are reported in Table 1. Concerning commercial CdS, the particle size reported in Table 1 was estimated considering the diffraction peaks attributed exclusively to the hexagonal phase (for example, 24.88, 28.26 and 36.74). When peaks that correspond to both structures (for example, 26.58 and 43.8) were considered, a size of 38–40 nm was estimated.

The SEM images in Figure 2 show that both CdS-HT and CdS-HTa400 are formed by assemblies of particles of few micrometers of size having round shape, nevertheless, the comparison with XRD results deserves a comment. In fact, Debye Scherrer calculation allows determining the size of the crystalline domains of the materials, which corresponds for all the samples to a few tens of nanometers, much less than the size of the particles observed in the SEM images. Then, it could be possible to consider the presence of some amorphous phase, which could increase the size of the particles without affecting the Debye Scherrer calculation, but the powder diffractograms allow excluding this possibility, as a very flat and clean baseline can be observed in Figure 1. Therefore, a possible explanation to this discrepancy is that the particles observed by SEM are actually formed by aggregates of nanometric crystals. During the annealing, the sample aggregation increases and also the micrometric aggregates observed by SEM appear larger and partially sintered.

The EDS patterns of CdS-HT, CdS-HTa400 and of commercial CdS reported in Figure 3 are very similar, and clear peaks of cadmium and sulfur are evident: CdS-HT was composed of 51.8% Cd element and 48.2% S element, and CdS-HTa400 was composed of 51.6% Cd and 48.4% S, while commercial CdS was composed of 51.7% Cd and 48.3% S. The atomic ratio is close to one, indicating that all samples are around the nominal composition.

Nitrogen adsorption/desorption isotherms of the investigated samples present the typical trend of low surface area materials, with evidence of a small hysteresis loop indicating the presence of mesopores (Appendix A). In agreement with the shape of the isotherms and as indicated by BET model applied to the experimental data, CdS-HT and CdS-HTa400 have a very low SSA (~5 m^2^/g for CdS-HT and ~3 m^2^/g for CdS-HTa400), consistent with their aggregated nature, but in both cases, higher than that of commercial CdS (~1 m^2^/g).

Concerning the porosity, studied through DFT method, both CdS-HT and CdS-HTa400 present a total amount of mesopores of 0.02 cm^3^/g, whereas the total volume observed for commercial CdS is almost negligible and equal to 0.01 cm^3^/g. The annealing seems to cause a sintering of the CdS-HT sample, as witnessed by the complete elimination of mesopores smaller than 100 Å of width during the thermal treatment (Figure 4), in agreement with the results obtained by XRD and SEM.

In XPS survey spectra of CdS-HT and of CdS-HTa400 (Appendix A), the main peaks were attributed to Cd 3d and S 2p transitions, with contributions from the C 1s, O 1s and Cl 2p signals (Appendix A). The large amount of C can be completely associated with adventitious carbon: in fact, the signal arose from the surface of the sample and was found almost identical in all materials, as expected for this type of surface contamination. Moreover, the eventual presence of C as dopant would be in a specific chemical state (usually a carbide peak in C 1s at 282–283 eV binding energy), as pointed out in Refs. [29,30]. The lack of this signal, and the presence of the one at 284.4 eV from adventitious carbon led us to rule out doping by carbon.

The high-resolution scans of Cd 3d and S 2p of CdS-HTa400 are shown in Figure 5. The binding energies for the Cd 3d5/2 and Cd 3d3/2 were observed at 405.6 eV and at 412.3 eV, respectively. S 2p 3/2 and S 2p 1/2 were found at 162.1 eV and 162.3 eV, respectively. The binding energy of S 2p and Cd 3d were in excellent agreement with previously reported findings [31]. Moreover, the chemical state of CdS was further confirmed by acquiring the Auger signal of Cd MNN (Kinetic Energy of Cd M_4_N_45_N_45_ was 381.0 eV) and calculating the Auger modified parameter (AP was 786.6 eV), which univocally corresponds to CdS (786.5 eV reported by Refs. [32,33]. The presence of CdO can be excluded since the relative signal of Cd 3d should appear at 404 eV. CdS-HT presents almost the same composition of CdS-HTa400, but unfortunately, the CdS-HT XPS spectrum presents some artifacts, usually called differential charging [34], that cannot be corrected after the acquisition: the Cd 3d and S 2p signal was broader than the CdS-HTa400 case but with no effective chemical shift observable, since the Auger modified parameter was unchanged (AP does not depend on calibration or absolute position of the peaks). Similar artifacts seem to be present also in the spectra reported by Balushi on similar materials [31] but with no further comments. The presence of such artifacts allows us to evaluate only the atomic chemical composition, with no further details about the possible chemical states different from CdS. CdS-HT has no Cl 2p signal, which can be explained by sublimation of the residual chloride contaminants during the thermal treatment.

In Figure 6 the absorption spectra of CdS-HT and of CdS-HTa400 powders are reported. It is observed that the absorption of the annealed sample extends to longer wavelengths with an absorption tail. This absorption does not appear in CdS-HT, which has a usual light-harvesting maximum efficiency in the range between 400 and 500 nm. An analogous behavior is observed for commercial CdS (Appendix A). The analysis of Tauc Plot graphs (Appendix A) allows to extract a direct E_g_ of 2.18 eV for CdS-HT, of 2.32 eV for CdS-HTa400 and of 2.31 for commercial CdS. Interestingly enough, in the case of CdS-HTa400, a second indirect band gap of 1.4 eV can be evaluated from extrapolation of the linear portion of the plot to the photon energy axis (Appendix A). This could be attributed to the presence of S vacancies or interstitial Cd or S that add intra band gap energy levels.

### 3.2. Photocatalytic Activity of CdS-HT and of CdS-HTa400

From the above characterization, the annealed material CdS-HTa400 is very similar to the starting CdS-HT from a morphologic/structural point of view, while the optical properties of the former are surprisingly different from those of the latter, as shown in Figure 6. In order to determine whether the extension of absorption at red wavelengths of CdS-HTa400 plays a role in its photocatalytic activity, we decided to use these materials for the reductive cleavage of N=N bond in methyl orange (MO), a test reaction already employed by some of us to study CdS-based photocatalytic materials [24].

In a typical experiment, the optimal quantity of CdS-HT (10 g/L) or CdS-HTa400 (3 g/L) was suspended in H_2_O/EtOH (10% *v*/*v*) reaction mixture containing MO (C_0_ = 10 ppm). After deaeration by N_2_ flux, the suspension was irradiated with visible light (λ > 400 nm). During the irradiation, the decreased intensity of the main ground state absorption of MO (λ max = 464 nm) was accompanied by the increase of a new absorption band (λ max = 245 nm), previously attributed to deprotonated sulfanilic acid, generated after the reductive cleavage of the azo bond in MO (Appendix A) [24]. Figure 7 reports the C/C_0_ ratio of MO as a function of the irradiation time using the above-mentioned photocatalysts. A very rapid decrease of methyl orange concentration is observed when CdS-HTa400 is used as a photocatalyst, and only after 10 min irradiation, the entire starting MO disappears from the solution (blue full triangles). By contrast, with CdS-HT the decrease of MO concentration is slower, and an almost complete disappearance of the dye is obtained after 60 min (red full triangles). A reasonable contribution to the increase in photocatalytic activity might be attributed to the slight increase in crystallinity following annealing (Figure 1) that could lead to more efficient separation of electron–hole pairs, decreasing the recombination process and allowing holes and electrons to undergo reduction and oxidation reactions on the surface of the photocatalyst in contact with the solution. However, the larger partly sintered aggregates imply the nearly halved surface area as observed by BET, and this could have a negative effect on the photocatalytic performance, since a smaller surface area corresponds to fewer surface active sites.

For comparison, we report the dye decay profile obtained with commercial CdS. Interestingly, CdS-HTa400 photoactivity is even greater than that of commercial CdS (black full circles) for which the total disappearance of MO occurs in a longer irradiation time (15 min). In contrast with hydrothermally prepared materials, the photocatalytic performance of commercial CdS does not improve when the material is previously heated at 400 °C (black empty circles). However, annealing at 400 °C of commercial CdS caused an analogous increase in crystallinity without any significant variation in the ratio between the cubic and hexagonal phases (Appendix A). Thus, the reason behind the superior performance of CdS-HTa400 with respect to commercial CdS remains elusive in these experiments. We believe that the profound reason for the great improvement in photocatalytic efficiency observed in the case of CdS-HTa400 and not observed for annealed commercial CdS is not to be found among the morphological and crystalline characteristics but rather in some intrinsically different electronic properties, resulting in distinct spectroscopic and electrochemical features.

Figure 7 also shows that neither irradiation of MO (green full squares) in the absence of any photocatalyst nor the initial suspension kept in the dark in the presence of CdS-HTa400 (empty blue triangles) led to a decrease in MO concentration, indicating that the simultaneous presence of visible light and photocatalyst is essential to drive the azo bond cleavage. 

In addition to the higher photocatalytic activity, CdS-HTa400 is also found to be recyclable for at least four repeated cycles with a final loss of activity of about 5% (Appendix A). In agreement with this result, the XPS spectrum of the material after photocatalysis shows no significant alterations except for a slight broadening of Cd and S signals (Appendix A). This is a relevant result in photocatalysis since CdS-HTa400 is an efficient, stable, and recyclable material overcoming all the photostability problems of commercial CdS, which, by contrast, exhibits a substantially decreased activity upon repeated photocatalytic cycles (Appendix A).

### 3.3. Spectroscopic Properties of CdS-HTa400

The photocatalytic performance of CdS-HTa400, which is higher than that of commercial CdS, cannot be attributed only to its crystallinity since, actually, larger coherent domains are found by XRD analysis in the latter (Figure 1). In addition, usually, the photoactive material that shows the simultaneous existence of multiple crystalline phases is more active since this feature favors electron/hole separation. In our study, however, CdS-HTa400 displays the only the hexagonal phase and is nevertheless more active than commercial CdS (either before or after 400 °C annealing), which has both the cubic and hexagonal phases.

Therefore, it becomes important to investigate and understand the electronic structure of the CdS-HTa400 material, as we believe it plays an important role in determining photocatalytic performance. In Figure 8, we report the formal quantum efficiency (FQE, i.e., number of dye molecules hydrogenated per one incident photon) vs. monochromatic irradiation wavelength for CdS-HTa400 and commercial CdS. It can be observed that CdS-HTa400 maintains a residual activity at wavelengths higher than 520 nm. In contrast, CdS is not active from 520 nm onward and furthermore its photoactivity is always lower than that of CdS-HTa400 throughout the wavelength range examined. Overall, we can conclude that the action spectra have a very similar shape, which correspond to the expected ground state absorption of CdS. The absorption tail, which, in CdS-HTa400, extends well over 800 nm, has no direct impact in extending the spectral sensitivity of the material for the reaction under study. The apparent slight red shift of the photoaction onset, found for CdS-HTa400, can be explained by its intrinsically higher FQE values. Nevertheless, the nearly doubled FQE in CdS-HTa400 compared with the commercial powder is a sure indication of either improved carrier separation in the former or of electron trapping at intrinsically more reactive sites in CdS-HTa400 following excitation, which allows azo dye reduction with higher efficiency or of both events.

When photoluminescence spectra were recorded (Figure 9), the slide of FTO/CdS-HTa400 exhibited an emission at around 520–530 nm and a more intense and broad tail extending into the infrared region. This last emission was not so evident in the case of FTO/CdS-HT. In agreement with previous studies [35,36,37] the emission at 520–530 nm can be attributed to electron–hole recombination (Equations (2) and (3)) and that at 700 nm onward to intrinsic defects such as sulfur vacancies (Vs), respectively. Electrons promoted into the conduction band, as a consequence of band gap excitation, are trapped in these vacancies (*V_S_*), which are believed to exist at about 0.7–0.8 eV below the conduction band (Equation (4) and Figure 2). This is also consistent with the gap extracted from the Tauc analysis of the low energy tail of CdS-HTa400 in Figure 6 and Appendix A. The radiative recombination of these trapped electrons with the holes photogenerated in the valence band corresponds to the observed red emission (Equation (5)). Concerning commercial CdS, we observe a red emission slightly less intense in comparison with CdS-HTa400, but an important emission relative to the band gap recombination (centered at 550 nm) is observed in this case, which suggests that the direct CB/VB radiative recombination pathway is still active in commercial CdS representing an additional energy wasting channel compared to CdS-HTa400.
(2)$$CdS+hv_{0}\to CdS\;\left(h_{VB}^{+}+e_{CB}^{-}\right)$$
(3)$$h_{VB}^{+}+e_{CB}^{-}\to hv_{1}$$
(4)$$V_{S\;}+e_{CB}^{-}\to V_{S}^{-}$$
(5)$$h_{VB}^{+}+V_{S}^{-}\to \;V_{S\;}+hv_{2}$$

Nevertheless, the results obtained from the comparative analysis of the emission spectra seem to indicate that intra band gap energy states due to *V_S_* are more significantly populated in CdS-HTa400, where they could act as a longer lived reservoir of electrons useful for the reduction process relevant to our investigation. The fact that CdS-HTa400 lacks the typical green emission of CdS (550 nm band), is indeed an indication of a competitively fast electron trapping into the high density of *V_S_* defects (Equation (4)), which are responsible for the low energy optical absorption described in Figure 6, practically quenching the direct radiative CB/VB recombination (Equation (2)).

We note that while electron excitation from these defects to conduction band states does contribute to light absorption in the CdS-HTa400, it is not expected to contribute to photocatalysis. Indeed, the oxidizing power of holes left within these defects is insufficient to intercept the hole scavenger, and a very short-lived charge separated state would be generated through this process. This explains why the action spectra of commercial CdS and CdS-HTa400 are remarkably similar in spectral extension, only differing in the intrinsic activity of the two different materials. We remark, however, that the rigorous comparison of emission intensities in solid films composed of aggregated particles must be made very cautiously, since, ultimately, the intensity of emitted light depends also on extrinsic factors such as light scattering, and non-radiative quenching within aggregates having different size distributions. Thus, this analysis can only have a qualitative validity and independent confirmation of our hypothesis has to be searched through the application of other electrochemical/photoelectrochemical methods (vide infra).

### 3.4. Electro and Photoelectrochemical Investigation of CdS-HTa400 and of CdS-HT

Dark cyclic voltammetry (0.2 V/−1 V vs. SCE) of FTO/CdS-HTa400, FTO/CdS-HT and FTO/CdS in aqueous solution containing sodium formate (1 M) is reported in Figure 10 (solid lines). In the case of FTO/CdS-HTa400 two waves are visible, a first less intense process at approximately at −0.6 V assigned to filling of the trap states generated by *V_S_*, followed at stronger negative polarizations by a second more intense process assigned to the reduction of a higher density of electron acceptor states, probably conduction band states. The same experiment done in the presence of dissolved methyl orange (MO, 10 ppm) shows the presence of the two previous processes with a slight shift of the first one to about −0.65/−0.7 V but the return wave is less intense, probably because some of the trapped electrons are consumed by the presence of the dye in solution (dashed blue line). Cyclic voltammetry on FTO/CdS (commercial) shows a behavior qualitatively similar to annealed material but the peaks are much less intense, demonstrating that a lower density of intra band gap states is present in this material (black solid line). In the presence of MO, the return wave shows a negligible peak (black dashed line). Moreover, the cyclic voltammetry of the FTO/CdS-HT electrode only shows the beginning of the second process around −1 V both in the absence and in the presence of MO (red lines) consistent with its poor reactivity in photocatalysis. In this case the lower crystallinity of CdS-HT may also play a role in lowering the electronic conduction of the film, resulting in the overall modest current recorded with this substrate. The conductivity of CdS, subject to different deposition conditions has been studied in detail [38], finding that an increase in CdS grain size is related to increased film conductivity. This is due to a lower density of grain boundaries, which constitute barriers for charge transfer across interconnected CdS domains. Furthermore, the improved crystallinity found upon annealing also results in the formation of spatially more extended lattice regions where the periodic Bloch conditions can be applied, resulting in the formation of delocalized bands with a low effective mass that are a necessary requirement for efficient charge transport [39,40].

The results reported in Figure 10 agree with photoluminescence spectra (Figure 9) and confirm the presence of intra band gap states, which act as electron traps. Moreover, we have direct evidence that the density of these intragap states is much superior in CdS-HTa400 and that these trapped electrons are transferred with a greater efficiency to MO, affording the photoconversion process that indeed proceeds faster in the case of CdS-HTa400.

Figure 11 reports chronopotentiometry experiments of FTO/CdS-HTa400, FTO/CdS-HT and FTO/CdS in deaerated solutions of sodium formate. It is observed that under illumination the photovoltage of FTO/CdS-HTa400 and of FTO/CdS is more negative (around −0.9 V vs. SCE) than that of FTO/CdS-HT (−0.6 V vs. SCE), indicating that the former two materials have a stronger reductive capability due to improved charge separation and that the quasi-Fermi level of FTO/CdS-HT is barely capable of intercepting the MO reduction, well explaining its modest photocatalytic performance. It is interesting to observe that when dark conditions are restored, the photopotential decay is very slow in the case of FTO/CdS-HTa400 while in the case of the other two CdS-based electrodes, the photopotential very rapidly shifts back to the initial values, losing most of the photovoltage amplitude within a few seconds. This result indicates that a significant density of sulfur vacancies, whose presence was first evidenced by optical methods are indeed capable of acting as a long lived electron reservoir: essentially, in the FTO/CdS-HTa400 case, when light is turned off, only a small fraction (ca 15%) of the total photovoltage amplitude is lost within a few seconds, whereas most of the decay recovers very slowly (tens to hundred seconds), according to a complex kinetics, consistent with a distribution of rate constants stemming from the high density of intragap states peculiar of this material. Noteworthy, after ca 500 s from the illumination cycle, the photovoltage is still of the order of −0.6 V vs. SCE.

By contrast, we observe that when light is switched on, the photovoltage increase is sudden, as in the other two CdS-based electrodes, showing that the population of these long-lived trap states is indeed fast. Thus, the CdS-HTa400 shows the type of kinetic asymmetry that is highly desirable in photocatalysis, fast charge separation but extremely slow recombination that allows the excess electronic charge trapped in reactive states, to be fruitfully employed for reductive interfacial charge transfer to MO type molecules or to trigger other kinds of kinetically demanding multielectron reductive chemistry [24].

## 4. Conclusions

In this study, the synthesis of a new CdS-based photocatalyst is accomplished via hydrothermal method. We demonstrate that post-synthesis annealing (CdS-HTa400) can determine crystal properties and spectroscopic characteristics and how their synergy affects the photocatalytic performance.

More in detail, upon annealing, there is the formation of sulfur vacancies, intra band gap states that act as an electron sink. CdS-HTa400 shows highly desirable characteristics in photocatalysis, such as fast charge separation but extremely slow recombination due to electron trapping in sulfur vacancies. This allows the excess electronic charge trapped in reactive states to be fruitfully employed for reductive interfacial multielectron transfer to methyl orange, which is totally transformed in ten minutes of visible irradiation.

In addition, CdS-HTa400 has a very high photostability, and it can be recycled for at least four consecutive runs with a loss of activity lower than 5%. These results are important if one compares CdS-HTa400 with commercial CdS, which is characterized by a similar photocatalytic activity, but it suffers from such low photostability that it cannot be recycled.

CdS-HTa400 represents a visible-light active photocatalyst that is overall recyclable, stable and more efficient than the commercial reference, opening up the development of a new visible-light reactive photocatalyst.

In this regard, we intend to investigate the effect of annealing temperature on material properties to maximally improve all the characteristics required to have a good photocatalyst.

## Data Availability

The data presented in this study are available from the corresponding author upon reasonable request.

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
