# Peer review of "A Novel Hydrothermal CdS with Enhanced Photocatalytic Activity and Photostability for Visible Light Hydrogenation of Azo Bond: Synthesis and Characterization"

_nanomaterials, 2023, doi:10.3390/nano13030413_

Round 1

Reviewer 1 Report

In this manuscript authors are investigating properties of the thermally treated (hydrothermally deposited) CdS versus untreated and commercial CdS respectively. It presents a series of investigation with some results (partial) interpretations, which in my opinion should be improved. Below are some punctual comments on some specific issues.

Introduction

“Noteworthy, hydrothermal CdS shows higher photocatalytic activity than commercial CdS powder”

- It should actually be useful for the readers to distinguish between the two with their properties rather then the deposition technique.

Hydrothermal Synthesis of Cadmium Sulfide

“For photoelectrochemical measurements fluorine doped tin oxide (FTO) was used as

ohmic support for hydrothermal CdS”

- What is the doping percent ?

Results and discussions:

Fig. 1: “XRD patterns of hydrothermal samples are different from that of commercial CdS which shows the presence of both hexagonal and cubic crystalline structures (Figure S1).”

- Why not presenting all patterns on the same image for comparison (after all they were probably recorded with the same tool and using same experimental parameters). Commenting the structural differences between the three spectra should be more simple an easier to follow by the readers.

Tab.1

- Since the commercial power have two types of structures, the estimated crystalite size belong to both types ? Is there any difference between the two crystalite size corresponding to the two structure types ?

“CdS-HT was composed of 51.8% Cd element and 48.2% S element, while CdS-HTa was composed of 51.6% Cd and 48.4% S”

- What is the ratio for commercial CdS ?

Fig. 6

Why it is not presented the results of the commercial power as well ?

Photocatalytic activity of CdS-HT and of CdS-HTa400

“From the above characterization, the annealed material CdS-HTa400 is very similar to the starting CdS-HT from a morphologic/structural point of view, displaying only a slightly higher crystallinity and larger partly sintered aggregates, which also imply, as a negative effect the nearly halved surface area observed by BET. However, the optical properties of CdS-HTa400 are surprisingly different from CdS-HT as shown in Figure 6”

- Why the increase of crystalinity and crystalite size accompanied by the Cd percent (probably from the crystalite boundaries) decrease is disregarded while interpreting these results ?

“We notice that the performance of commercial CdS does not improve after heat treatment at 400°C (circles) while annealing of CdS-HT plays an important role in generating a photocatalyst which is itself very active and more active than the commercial benchmark (triangles)”

- the authors are not characterizing in any way the 4000 C treated commercial CdS and consequently they are not trying to understand the effect of their treatments and processes. Same observation for Fig. 9 !

Fig. 8

-Why the untreated hydrothermal CdS curve is not presented there ?

“In this case the lower crystallinity of CdS-HT may also play in role in lowering the electronic conduction of the film, resulting in the overall modest current recorded with this substrate”

- Could authors develop this idea ? Is the crystalite size smaller than the one of the commercial CdS ? How about the Cd ration decrease influence ?

“This result indicates that a significant density of sulfur vacancies, whose presence was first evidenced by optical methods are indeed capable of acting as a long lived electron reservoir: essentially, in the FTO/CdS-HTa400 case”

- Actually S vacancies should be less in the thermally treated CdS according to authors previous investigations. Any comments on that ?

Reviewer 2 Report

This work deals with the preparation of CdS, via a solvothermal procedure followed by annealing of the so obtained solid. The work deals also with the study of its photocatalytic activity in the reduction of MO as a probe which compared with both a commercial CdS and the unannealed CdS. The methodologies for characterization and the photocatalytic activities of the samples are suitable to the purposse of the study. The results strongly support the superior activity of the annealed catalyst, as is hypotesized by the authors,  is probably due to the presence of vacancies that favour both stabilizing of the separated charges under radiation and transference of electrons to the substrate.

However, relative to the above point the nature of C component existing in the samples deserves a tentative explanation; specially in the case of the annealed catalysts the probable existence of C as dopant (which, according to Figure S3 could has been formed under annealing), coming from surfactants linked to unannealed CdS, (vide infra) should be discussed. It has been pointed out that metallic photocatalysts dopped with C (BiOX case) strongly enhanced the photocatalytic activity.

In addition to the above one, the following points should be revised before publication of this paper:   

* In subsection 3.1. 

According to the data of the paragraph in the row 204 of the manuscript it must be written "....seven...." instead "six"

* Regarding the discussion of paragraph of rows 224-236, probably HRTEM images (if possible although not necessarily) would be useful to prove the crystal nature of aggregates in Figure 1to which the discussion is referred to. To the extend that a close relationship is expected between the catalytic activity of the CdS materials and their crystal stuctures I would recommend to do the above if possible.

*In the caption of Table S1 it should be indicated the atom composition units ( %?)

* Unless I had missunderstood, the significant amounts of C component of CdS-HT and commercial CdS don´t fit the comment of row 266 of the manuscript about the contribution of this component is minor in CdS-HT and the commercial CdS. On the contrary the corresponding amounts c.a. 45%,  (Table S1) are important and this fact deserves a tentative explanation. Where could this component comes from?.

*The component at higher B.E. of S signal in XPS spectrum of CdS-HT (Figure 5b) should be assigned too. 

In my oppinion, the comparison of the absorbances (Figure 6) of CdS-HT and CdS-HT-a400, as done in the paragraphs  of 288-296 and 300-308, would require the  concentration of the catalysts in BaSO4 pellets are similar. This is an important issue that has been not indicated in the Experimental Section. On the other side the shape of the absorption curves of both catalysts in the 500-700 nm range indicates an absorption tail in CdS-HT which is absent in the annealed catalyst. In spite of it is not clearly observed in the corresponding TAUC plot this could be due to an additional transition . I suggests to discusse any possible connection of this fact with the large amount of C in CdS-HT which should come from thiourea and/or Cysteine coordinated to Cd centres.  The possibility of appearing of some residual C on the sample after annealing at 400º and possible influence in the photocatalytic activity  of CdS-HTa400 (see e.g. J. Di, J. Xia, H. Li, S. Guo S. Dai, Nano Energy 41 (2017) 172-192) should be discussed too. 

Reviewer 3 Report

The article can be published after a minor revision.

The problems that need to be corrected/completed are the following:

- correctly writing Debye Scherrer's equation;

- specifying the value of  λ.

- please check the title placed on the OY axis, in figure 1;
- please ensure better clarity/visibility for figure 7;
- the Conclusions subsection should be completed with concrete, numerical results.

Round 2

Reviewer 1 Report

The authors have improved the manuscript and is easier to follow by the reader, particularly by including wherever was possible commercial CdS material parameters. However, I still have a remark about Fig. 7. While commenting about presenting results of the 4000C annealed commercial CdS and as purchased, on one hand, and 4000C annealed and non-annealed hydrothermal CdS material without actually trying to characterize the effect of the annealing over the commercial powder versus the annealing effect over the hydrothermal CdS the authors replay was:

The point is that Figure 7 demonstrates that thermal treatment at 400°C of commercial CdS does not induce differences in the photocatalytic performance. This is in contrast to what was observed with hydrothermally prepared materials”

In my opinion, presenting a result of an annealing process, over a material versus another material, without trying to understand ‘Why is that happening ?” is not a very “scientific approach”, and, instead of “clarifying” some aspects to the reader, it might actually be confusing the reader. This could actually make him “associate” some material property changes with the “preparation method” or “treatment method” rather then with the (obtained) material specific properties and characteristics, and in my opinion this could be rather ‘misleading’ for the reader. If the increase of the catalytic efficiency is because: “The higher crystallinity of CdS-HTa400 (as reported in Figure 1) with respect to CdS-HT could lead to more efficient separation of electron-hole pairs with respect to the latter, decreasing the recombination process and allowing holes and electrons to undergo reduction and oxidation reactions on the surface of the photocatalyst in contact with the solution, […] “, then the authors should show that the thermal treatment of the commercial CdS does not change that. Or find a another way to support their results, if this is not the case.
